# Graph Autoencoder-based Motif Extraction Algorithm for Molecular Representation Learning

## Abstract

Molecular representation learning using graph neural networks(GNNs) has become a research hotspot in the fields of chemistry and biology in recent years. The pretraining-finetuning paradigm has been widely used to address the issue of limited labeled molecular datasets, achieving great success due to its ability to leverage large amounts of unlabeled data. Additionally, frequently occurring molecular substructures, known as motifs, can often capture the local information and higher-order connectivity of molecules more effectively, providing a better paradigm for pretraining. However, existing motif extraction methods face the issues of relying on domain-specific knowledge and neglecting the local structural information of atoms. To address these problems, we propose a motif self-extraction method based on a graph autoencoder. This method utilizes the graph autoencoder for structural reconstruction, allowing the model to automatically identify frequently occurring local patterns. Furthermore, we also propose a motif-based pretraining method that simultaneously captures the local information and higher-order connections of both the molecular graph and the motif graph. We pretrain on the 250K Zinc15 dataset and conduct downstream performance prediction on eight commonly used molecular property prediction datasets. Experimental results demonstrate the effectiveness of our method.

## 1 Introduction

Molecular representation learning (MRL) has garnered increasing attention in recent years in the fields of chemical analysis and drug discovery research. Molecules can be naturally represented as graphs, making graph neural networks a popular choice for molecular analysis. However, while vast amounts of unlabeled molecular data are available, labeled datasets remain scarce. Inspired by the success of the pretraining-finetuning paradigm in the fields of computer vision (CV) and natural language processing (NLP), recent studiesHu et al. (2019); Wang et al. (2022); Zhang et al. (2021) have explored pretraining models on large unlabeled molecular datasets before fine-tuning them for specific downstream tasks. For example, GROVERRong et al. (2020) utilize large-scale unlabeled datasets to generate chemical and topological labels for molecular predicting learning, AttrMaskingHu et al. (2019) uses node embeddings to predict attributes of masked nodes. The generative method Hu et al. (2020); Zhang et al. (2021) learns the distribution of molecular graphs by training a model. Contrastive learning Wang et al. (2022); Stärk et al. (2022); Xiang et al. (2023); Luong & Singh (2024), as a popular approach in unsupervised methods, is also widely applied in molecular pretraining. InfoGraphSun et al. (2019) maximizes the mutual information between the graph-level representation and substructure representations at different scales to learn the graph-level representation. GraphMVPLiu et al. (2021) introduced 3D molecular graphs, leveraging contrastive learning between 2D-graphs and 3D-graphs. Although the model's performance improved, the cost of obtaining 3D-graph coordinates is high.

Typically, most work focuses on pretraining at the node or graph level. Node-level pretraining captures the local structure of molecules but overlooks higher-order structural arrangements, while graph-level pretraining may neglect more fine-grained details within the molecules. To address this problem, some studies have proposed motif-based pretraining methods Zhang et al. (2021); Yan et al. (2024); Luong & Singh (2024); Zang et al. (2023), where a motif is defined as a pattern

Table 1: Compare existing motif extraction methods and the problems they face. *No Over-Frag.* : molecule is not overly decomposed into too many small motifs; *High-Freq.* : extracted motifs are high-frequency; *Knowledge-Free* : no expert knowledge is required for guidance; *Considers Local-Struc.* : local structural information of the atomic is considered during motif extraction.

| Method | No Over-Frag. | High-Freq. | Knowledge-Free | Considers Local-Struc. |
|---|---|---|---|---|
| BRICSDegen et al. (2008) | ✓ | - | - | - |
| JT-VAEJin et al. (2018) | - | ✓ | - | - |
| MGSSLZhang et al. (2021) | - | ✓ | - | - |
| DBPGYan et al. (2024) | ✓ | - | ✓ | - |
| PS-VAEKong et al. (2022) | ✓ | ✓ | ✓ | - |
| OURS | ✓ | ✓ | ✓ | ✓ |

of connections in a complex network that appears significantly more frequently than in a random network Milo et al. (2002). Motif-based pretraining methods require first using motif extraction techniques to convert molecular graphs into motif graphs. Inspired by chemistry, BRICSDegen et al. (2008) utilizes domain-specific knowledge related to molecules to customize 16 rules for motif decomposition of molecules; however, the extracted vocabulary is often quite large and contains many specific or low-frequency fragments which poses significant challenges for motif recognition. To overcome these shortcomings, JT-VAEJin et al. (2018) further decomposes molecules at the ring and bond levels, while MGSSLZhang et al. (2021) customizes two new rules to further segment the generated motifs. However, reducing the fragment size hinders the ability to capture higher-order representations of molecules, and these methods all require domain-specific knowledge for guidance. Although DBPGYan et al. (2024) and PS-VAEKong et al. (2022) proposed automatically extracting motifs based on structural similarity and frequency, respectively, they only consider atomic bonds and their terminal atoms when capturing molecular motifs, while ignoring the local structural information of atoms, which is clearly unreasonable.

To address the aforementioned problems, we propose a graph autoencoder-based motif extraction method (GAME) in Section 3.1. Specifically, we first train a graph autoencoder to perform structural reconstruction on a large-scale unlabeled molecular dataset. When two atoms are connected, the model will make their representation, which include local structural information, more similar to each other. After multiple optimizations, the representations of frequently occurring connection patterns become more and more similar. We then input the molecules into the pre-trained graph autoencoder and evaluate the connection frequencies between all atom pairs. By combining this information with the molecular structure graph and removing low-frequency atomic bonds, we decompose the molecules into multiple motifs. After processing a large number of unlabeled molecules, we filter out motifs with higher frequencies to form a motif vocabulary, which can be used to extract motif graphs from unseen molecules. Table 1 compares all the motif extraction methods and the problems they encounter. Although previous methods address several of these problems, they still have limitations.

In addition, we propose a motif-based molecular pretraining method that captures both the local structures and higher-order connectivity of molecules. The model is mainly divided into three parts: (1) motif-based contrastive learning, which performs contrastive learning at the motif level to enhance the interaction of information between atoms and motifs; (2) cross-level matching learning, which establishes a matching task that connects motif-level and molecular-level training; (3) prototype-based motif prediction learning, which uses prototype representations for motif classification to alleviate the class imbalance and overfitting problems encountered by previous prediction methods.

The major contributions of this work are as follows:

- We identify limitations in existing motif extraction methods and propose GAME, a graph autoencoder-based approach that captures local atomic structures without requiring domain-specific knowledge. To our knowledge, this is the first GNN-based motif extraction method.

- We introduce a motif-based molecular pretraining framework MBMP that jointly leverages molecular and motif graphs to learn both local features and high-order connectivity.

- Extensive experiments on eight molecular property prediction datasets show that our method consistently outperforms strong baselines, demonstrating its effectiveness and generalizability.

- We further validate our approach through comparative studies with existing motif extraction and motif-based pretraining methods, demonstrate the effectiveness of the proposed two methods.

## 2 RELATED WORK

### 2.1 MOLECULAR REPRESENTATION LEARNING

Traditional approaches Carhart et al. (1985); Nilakantan et al. (1987); Rogers & Hahn (2010) commonly represent molecular structures through fingerprints. Earlier works Svetnik et al. (2004); Wu et al. (2018); Meyer et al. (2019) utilized tree-based models such as Random Forest Breiman (2001) and XGBoost Chen & Guestrin (2016) to predict molecular properties using these fingerprints. String-based representations Shen & Nicolaou (2019); Yüksel et al. (2023) generally use SMILES and InChI notations as inputs, leveraging the language modeling capabilities to extract features. On the other hand, graph-based representations Gilmer et al. (2017); Zhang et al. (2021); Chen et al. (2024) treat atoms as nodes and bonds as edges, utilizing graph neural networks (GNNs) to capture molecular structure. Recently, unsupervised pretraining Wang et al. (2019); Li et al. (2021); Wang et al. (2022); Stärk et al. (2022) has been increasingly applied to tackle the challenge of limited labeled data. For example, SMILES-BERT Wang et al. (2019) uses transformer layers with attention and is pre-trained on 35 million compounds from ZINC15 through a masked SMILES recovery task. MPG Li et al. (2021) and GROVER Rong et al. (2020) applies a graph-based framework combined with effective node and graph-level pretraining strategies for learning molecular representations. MoleculeSTM Liu et al. (2023) integrates contrastive learning to simultaneously learn molecular chemical structures and their textual descriptions. Git-Mol Liu et al. (2024) extends this approach by incorporating a multimodal large language model that integrates graph, image, and text data to capture the intricate details of molecular structures and their corresponding images. Despite their advancements, these methods primarily target molecular or atomic-level representations.

### 2.2 MOTIF EXTRACTION ALGORITHM

Recently, motif-based pre-training methodsZhang et al. (2020; 2021); Zang et al. (2023); Luong & Singh (2024) have been proposed to better extract molecular features using motifs or fragments of molecules. The success of these methods relies on the development of motif extraction techniques. Early approachesDegen et al. (2008) used predefined chemical rules to fragment molecular bonds and derive motifs, but this often led to numerous low-frequency motif fragments. To address this issue, Jin et al. (2018) and Zhang et al. (2021) further refined the motifs generated by BRICS, producing smaller yet more frequent fragments. Furthermore, in an effort to eliminate reliance on expert knowledge, Kong et al. (2022) and Yan et al. (2024) proposed self-supervised motif extraction methods that iteratively capture frequently co-occurring atoms from large datasets to form new nodes, ultimately yielding larger and more frequent motifs. However, these methods fail to consider the local influence of atoms on motif formation.

## 3 METHODOLOGY

### 3.1 GRAPH AUTOENCODER-BASED MOTIF EXTRACTION

Unlike traditional motif extraction methods, we propose an innovative algorithm that eliminates the reliance on specialized knowledge while fully leveraging the local structural information of molecules. Our motif extraction framework consists of two key stages: (1) training a graph autoencoder for capturing high-frequency connection patterns and (2) discovering motifs using the trained graph autoencoder. In the following sections, we will provide a detailed explanation of each stage, including its application and underlying principles.

#### 3.1.1 TRAINING A GRAPH AUTOENCODER.

Among the existing motif extraction algorithmsDegen et al. (2008); Jin et al. (2018); Zhang et al. (2021); Kong et al. (2022); Yan et al. (2024), only Kong et al. (2022) and Yan et al. (2024) do not rely on expert knowledge. However, when partitioning motifs, they only consider the bond

between atoms and the relationship between the atoms at both ends, without fully taking into account the local structural information of each atom. This is clearly limited, as whether two atoms belong to the same motif is influenced not only by their bonding relationship but also by their local environment, such as whether they are part of a ring or how strongly they are connected to other atoms.

**Question 1: How can we effectively capture the local structural information of atoms?** To capture the local structural information, a common approach is to represent the molecule as a graph, with atoms as nodes and bonds as edges. Graph neural networks are then employed to aggregate information between nodes, effectively capturing the local structural features of atoms. However, while graph neural networks can capture local information, they are not able to directly capture the high-frequency connectivity patterns of molecules.

**Question 2: How can GNNs be used to capture the high-frequency connectivity patterns of molecules?** Inspired by Kipf & Welling (2016) where a graph autoencoder is used for structural reconstruction to optimize the model, we found that extending this approach to molecular graphs allows for the capture of high-frequency connectivity patterns. Specifically, when a large number of molecular graphs are fed into the autoencoder for structural reconstruction, frequent patterns of connectivity (e.g., $-C = O$ and $-CH_3$) will lead to repeated adjustments during optimization, making their embeddings more similar. After multiple rounds of optimization, the model will naturally tend to bring the embeddings of atom pairs with such local connectivity patterns closer together. In contrast, for connectivity patterns that occur infrequently or not at all, the model will reduce the similarity of the corresponding atom pair embeddings. After training, the model will have the ability to capture high-frequency connectivity patterns, which can then be used for motif extraction.

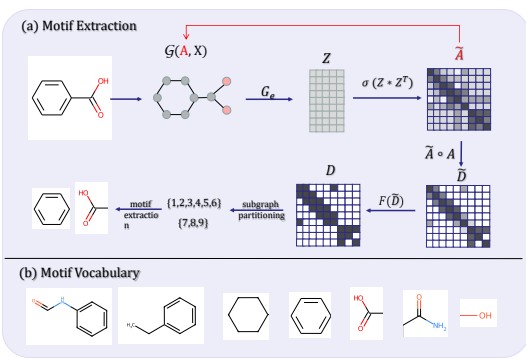

Figure 1: The framework of the motif extraction algorithm.

The specific training process is shown in Figure 1. For a molecule $\mathcal{M}$, we first represent it as a graph $\mathcal{G}(\mathcal{V}, \mathcal{E}) = (A, X)$, with atoms as nodes and bonds as edges. This graph is then passed through a graph encoder $G_e$, which uses its aggregation mechanism to capture the local structural patterns of the atoms:

$$Z = G_e(A, X) \tag{1}$$

After obtaining the embeddings for each atom, we use a decoder to calculate the probability of an edge existing between each pair of atoms:

$$\tilde{A} = \sigma(ZZ^T) \tag{2}$$

where $\sigma$ represents the sigmoid function. Finally, we use binary cross-entropy loss $\mathcal{L}_B$ to reconstruct the molecular structure:

$$\mathcal{L}_B = -\sum_j \sum_k a_{jk} \log \tilde{a}_{jk} + (1 - a_{jk}) \log(1 - \tilde{a}_{jk}) \tag{3}$$

where $a_{jk}$ and $\tilde{a}_{jk}$ are the element in the $j$-th row and $k$-th column of $A$ and $\tilde{A}$, respectively.

### 3.1.2 MOTIF DISCOVER THROUGH THE TRAINED GRAPH AUTOENCODER

After several optimization iterations, the graph autoencoder effectively captures the high-frequency connectivity patterns of the molecule. When a new molecule is fed into the autoencoder, it can assess the frequency of the connection between two atoms within their local structural environment. However, we cannot directly define molecular motifs based solely on connection frequency, as some high-frequency connections may not correspond to actual bonds in the molecule. Therefore, it is essential to analyze this in conjunction with the molecular graph structure.

**Question 3: How can we use $A$ and $\tilde{A}$ for motif extraction in molecules?** Typically, motif extraction is achieved by determining whether atomic bonds should be broken, ultimately resulting

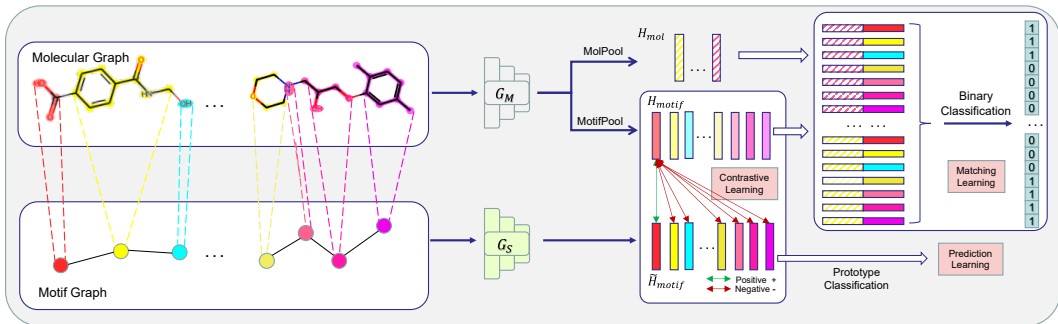

Figure 2: The overall framework of Motif-based Molecular Pretraining.

in multiple subgraphs. To achieve subgraph partitioning, we first remove edges that do not actually exist:

$$\tilde{D} = \tilde{A} \circ A \tag{4}$$

where $\circ$ represents Hadamard (element-wise) product. $\tilde{D}$ sets the positions where there are no edges in $\tilde{A}$ to 0, while keeping the positions with edges unchanged. To further remove low-frequency connectivity patterns, we define a function $F(X)$:

$$F(X) = \begin{cases} 1, & if \quad X_{ij} \geq k \\ 0, & if \quad X_{ij} < k \end{cases} \tag{5}$$

where $k$ is a hyperparameter used to determine whether the atomic bond is broken, $X_{ij}$ is the element in the $i$-th row and $j$-th column of $X$. By applying $D = F(\tilde{D})$, we filter out low-frequency edges, which typically represent the breakpoints of motifs. As a result, the molecule is divided into several smaller subgraphs. We collect the indices of all subgraphs and identify these substructures within the molecular graph, treating each substructure as a motif.

We trained on the 250K Zinc15 dataset and collected the motif sets of its molecules. To ensure the motifs are of high frequency, we further split those motifs with a frequency lower than the hyperparameter $N$, resulting in a high-frequency motif vocabulary $\mathbb{V}$.

After obtaining the motif vocabulary, we constructed the corresponding motif graph for each molecule. For a molecular graph $\mathcal{G}(\mathcal{V}, \mathcal{E}) \in \mathcal{X}_{\mathcal{G}} = \{\mathcal{G}_1(\mathcal{V}_1, \mathcal{E}_1), \mathcal{G}_2(\mathcal{V}_2, \mathcal{E}_2), \cdots, \mathcal{G}_n(\mathcal{V}_n, \mathcal{E}_n)\}$ and $n$ is the number of molecule, let $M = \{S^{(0)}, S^{(1)}, \cdots, S^{(m)}\}$ be its corresponding motif, where $S^{(i)}(\tilde{\mathcal{V}}^{(i)}, \tilde{\mathcal{E}}^{(i)}) \in \mathbb{V}$ is a subgraph,$\tilde{\mathcal{V}}^{(i)} \cap \tilde{\mathcal{V}}^{(j)} = \emptyset$ and $\cup_{i=1}^{m} \tilde{\mathcal{V}}^{(i)} = \mathcal{V}$. We denote the motif graph of $\mathcal{G}$ as $\tilde{\mathcal{G}}(\tilde{\mathcal{V}}, \tilde{\mathcal{E}})$, where $|\tilde{\mathcal{V}}| = |F|$ and each node $v_F^{(i)} \in \tilde{\mathcal{V}}$ corresponds to a motif $S^{(i)}$. If there exists at least one edge connecting the atoms of two motifs, an edge is established between the corresponding nodes of these motifs, defined as $\tilde{\mathcal{E}} = \{(i,j) | \exists u, v, u \in \tilde{\mathcal{V}}^{(i)}, v \in \tilde{\mathcal{V}}^{(j)}, (u, v) \in \mathcal{E}\}$. For simplicity, we retained the edge features within each motif, and the node features of the motifs are derived from embeddings in an optimizable lookup table.

### 3.2 MOTIF-BASED MULTI-LEVEL MOLECULAR PRETRAINING

In this section, we propose a motif-based multi-level molecular pretraining(MBMP), which consists of three components: motif-based contrastive learning, cross-level matching learning, and prototype-based motif prediction learning. Figure 2 illustrates the overall framework of the pretraining.

#### 3.2.1 MOTIF-BASED CONTRASTIVE LEARNING

We first define two graph neural networks, $G_M$ and $G_S$, to encode the molecular graph and the motif graph, respectively. Given a molecular graph $\mathcal{G}(\mathcal{V}, \mathcal{E})$ and its corresponding motif graph $\tilde{\mathcal{G}}(\tilde{\mathcal{V}}, \tilde{\mathcal{E}})$, we obtain their node representations separately as follows:

$$H_{atom}^{(i)} = G_M(\mathcal{V}, \mathcal{E}) \tag{6}$$

$$\tilde{H}^{(i)}_{motif} = G_S(\tilde{\mathcal{V}}, \tilde{\mathcal{E}}) \tag{7}$$

where $H^{(i)}_{atom}$ and $H^{(i)}_{motif}$ represent the embeddings of the molecule's atoms and motifs, respectively. After obtaining the representations of atoms and motifs, we perform contrastive learning at the motif level to better capture the molecule's local structural information. Contrastive learning optimizes the model by pulling positive samples closer together while pushing negative samples apart in the embedding space. In the context of unlabeled data, this typically involves generating different views of the same sample, treating data from the same sample as positive samples and data from other samples as negative samples. Following this approach, we define a MOTIFPOOL function to perform average pooling on the atomic embeddings belonging to the same motif:

$$H^{(i)}_{motif} = \text{MOTIFPOOL}(H^{(i)}_{atom}, \text{MAP}(\mathcal{V}, \tilde{\mathcal{V}})) \tag{8}$$

where $\text{MAP}(\cdot)$ is a mapping function that establishes the relationship between motifs and atoms. We then minimize the motif-based contrastive learning objective using the InfoNCE loss:

$$\mathcal{L}_{\mathcal{C}} = -\log \frac{exp\left(\left\langle H^{(i)}_{motif,r}, \tilde{H}^{(i)}_{motif,r}\right\rangle\right)}{\sum_{j=1}^{n}\sum_{k=1}^{|\tilde{\mathcal{V}}_j|} exp\left(\left\langle H^{(i)}_{motif,r}, \tilde{H}^{(j)}_{motif,k}\right\rangle\right)} \tag{9}$$

where $H^{(i)}_{motif,r}$ and $\tilde{H}^{(i)}_{motif,r}$ are the $r$-th row of $H^{(i)}_{motif}$ and $\tilde{H}^{(i)}_{motif}$, respectively. $|\tilde{\mathcal{V}}_j|$ is the number of motifs in the $j$-th molecule.

### 3.2.2 CROSS-LEVEL MATCHING LEARNING

While motif-based contrastive learning enhances information exchange between atomic and motif-level data, it overlooks the interaction between motif-level and molecular-level information. To address this limitation, we propose a cross-level matching learning approach. We aggregate the atomic node representations to generate the embedding for the entire graph:

$$H^{(i)}_{mol} = \text{MOLPOOL}\left(H^{(i)}_{atom}\right) \tag{10}$$

where $\text{MOLPOOL}(\cdot)$ represents the average pooling operation for atom belonging to same molecule. Through permutation and concatenation, we iteratively combine molecular and motif representations, and train a discriminator to predict the existence of a relationship. Given a permutation $\mathcal{P}$, the corresponding matching labels $y = \{y_{1,1}, \ldots, y_{1,N}, y_{2,1}, \ldots, y_{M,N}\}$ and the discriminator $d$, the matching loss is:

$$\mathcal{L}_{\mathcal{M}} = -\frac{1}{|\mathcal{P}|}\sum_{(i,j)\in\mathcal{P}} y_{i,j}\log\left(d\left(\tilde{H}^{(i)}_{motif}, H^{(j)}_{mol}\right)\right) \tag{11}$$

$$+ (1 - y_{i,j})\log\left(1 - d\left(\tilde{H}^{(i)}_{motif}, H^{(j)}_{mol}\right)\right) \tag{12}$$

By establishing cross-level matching learning, the training process encourages information to interact between the two levels.

### 3.2.3 PROTOTYPE-BASED MOTIF PREDICTION LEARNING

Unlike previous methods that directly predict motif labels, which may suffer from motif class imbalance and overfitting, we leverage prototype representations to enhance the predictive learning of motifs. Specially, for the $k$-th motif class, $\mathcal{C}_k$, we first obtain its class prototype embedding $p_k$:

$$p_k = \frac{1}{|\mathcal{C}_k|}\sum_{j\in\mathcal{C}_k} \tilde{H}_{motif,j} \tag{13}$$

where $|\mathcal{C}_k|$ is the number of motifs in class $k$ and $\tilde{H}_{motif,j}$ is the embedding of the $j$-th motif. We use prototypes for motif classification, assigning each motif to the nearest class prototype. To achieve this, we minimize the distance between the motif and its corresponding class prototype, while maximizing the distance to other class prototypes. The loss function $\mathcal{L}_{\mathcal{P}}$ is used for this classification task:

$$\mathcal{L}_{\mathcal{P}} = \frac{1}{M}\sum_{i=1}^{|\mathbb{V}|}\sum_{k=1}^{n_i} -\log \frac{\exp\left(-\|\boldsymbol{z}_k - \boldsymbol{p}_i\|^2\right)}{\sum_{i'\in\mathcal{C}_i}\exp\left(-\|\boldsymbol{z}_k - \boldsymbol{p}_{i'}\|^2\right)}, \tag{14}$$

where $|\mathbb{V}|$ is the number of motif class, $M$ is the total number of motifs, and $n_i$ represents the motifs number of class $i$.

Finally, we combine these three training methods. Since contrastive learning and matching learning involve optimizing two models, while predictive learning involves only motif model, we use $\alpha$ as a weight hyperparameter. The joint pretraining objective is then:

$$\mathcal{L} = \alpha\mathcal{L}_{\mathcal{P}} + (1 - \alpha)(\mathcal{L}_{\mathcal{C}} + \mathcal{L}_{\mathcal{M}}) \tag{15}$$

After pre-training, we obtained two models $G_M$ and $G_S$. Inspire by Luong & Singh (2024), we using the embeddings generated $G_M$ and $G_S$, concatenating them for downstream fine-tuning tasks.

## 4 EXPERIMENT

### 4.1 EXPERIMENTAL SETTINGS

**Datasets.** We use a processed subset containing 250k unlabeled molecules sampled from the ZINC15 databaseIrwin et al. (2012) for pretraining. As shown in Section 3.1, we first extract a motif vocabulary. To ensure proper motif division for unseen molecules, we expand the vocabulary by adding atoms that are not present in the pretraining dataset. In the pretraining phase, we generate motif graphs for the ZINC15 dataset using the motif vocabulary and perform motif-based molecular pretraining. For downstream tasks, we evaluate our method on eight binary graph classification tasks from MoleculeNet Wu et al. (2018). To simulate real-world applications, we apply scaffold splitting to partition the downstream dataset, dividing the data into training, validation, and test sets with an 80%:10%:10% split based on molecular structure. We conduct three independent runs for each data split and report the mean and standard deviation.

**Model Configuration.** In the GNN-based motif extraction algorithm, we employ a two-layer GIN (Graph Isomorphism Network) as the graph encoder. For molecular pretraining, we adopt a 5-layer GIN as the molecular encoder and a shallower 2-layer GIN for motif encoding, both with hidden layer dimensions of 300 as in previous workLuong & Singh (2024); Zhang et al. (2021). During the downstream fine-tuning phase, we append a MLP to the pre-trained model for classification.

**Baselines** As shown in Table 2, we compare our model with several notable pre-trained baselines in the molecular classification task, including predictive methods (AttrMask & ContextPredHu et al. (2019) , G-Motif & G-Contextual Rong et al. (2020) ), generative method (GPTGNNHu et al. (2020)), contrastive methods (GraphLoG Xu et al. (2021), GraphCL You et al. (2020), JOAOYou et al. (2021), JOAOv2You et al. (2021), GraphMVPLiu et al. (2021)), and motif-based method(MGSSLZhang et al. (2021), GraphFPLuong & Singh (2024), DGPMYan et al. (2024)).

**Implementation details.** In pretraing stage, we use the Adam optimizer to train ours framework with initial learning rate 1e-3, epoch 100, and batch size 32. We reduce the learning rate by a factor of 0.1 every 5 epochs without improvement. We use the models at the last pretraining epoch for fine-tuning. In the fine-tuning stage, to ensure comparability, our setting is mostly similar to that of previous works Liu et al. (2021); Luong & Singh (2024): Adam optimizer, initial learning rate chosen from {1e-3, 1e-4}, epoch 100, batch size 32, and dropout rate chosen from {0.0, 0.5}. We reduce the learning rate by a factor of 0.3 every 30 epochs.

### 4.2 RESULTS ON DOWNSTREAM TASKS

Table 2 report the results on 8 molecular property prediction benchmark. To provide a thorough evaluation, we compare MBMP with existing molecular pretraining methods and introduce four ablated variants: 1) without $G_S$ in downstream evaluation (w.o. S); 2) without contrastive learning (w.o. C); 4) without matching-based learning (w.o. M); 4) without predictive learning (w.o. P). The experimental results are shown in Table 2, where we summarize all methods into five different categories. The results show that our method consistently outperforms prior approaches, with an average ROC-AUC gain of 1.66% over the best baseline, and achieves state-of-the-art performance on 6 out of 8 datasets. Furthermore, the ablation study on MBMP and its variants highlights the importance of each pretraining component. Among the three strategies, contrastive learning contributes the most to performance, followed by matching learning and predictive learning. Notably, utilizing

Table 2: Test ROC-AUC on eight molecular property prediction datasets. The mean and standard deviation are reported for five random seeds. The top-3 performances on each dataset are shown in red color, with **red** being the best result, red being the second best result, and red being the third best result. We summarize the methods into 5 categories: (1) No-pretrain; (2) prediction or generation learning methods ; (3) contrastive learning methods ; (4) motif-based pretraining methods ; (5) MBMP and its variants.

| Method | BBBP | Tox21 | Toxcast | SIDER | ClinTox | MUV | HIV | BACE | Avg.AUC |
|---|---|---|---|---|---|---|---|---|---|
| No-pretrain | 65.6±1.4 | 71.5±1.0 | 61.5±0.8 | 59.4±1.2 | 66.5±5.2 | 74.5±0.5 | 64.4±1.9 | 72.6±1.9 | 67.00 |
| AttrMaskingHu et al. (2019) | 64.3±2.8 | 76.7±0.4 | 64.2±0.5 | 61.0±0.7 | 71.8±4.1 | 74.7±1.4 | 77.2±1.1 | 79.3±1.6 | 71.15 |
| ContextPredHu et al. (2019) | 68.0±2.0 | 75.7±0.7 | 63.9±0.6 | 60.9±0.6 | 65.9±3.8 | 75.8±1.7 | 77.3±1.0 | 79.6±1.2 | 70.89 |
| G-MotifRong et al. (2020) | 66.9±3.1 | 73.6±0.7 | 62.3±0.6 | 61.0±1.5 | 77.7±2.7 | 73.0±1.8 | 73.8±1.2 | 73.0±3.3 | 70.16 |
| G-ContextualRong et al. (2020) | 69.9±2.1 | 75.0±0.6 | 62.8±0.7 | 58.7±1.0 | 60.6±5.2 | 72.1±0.7 | 76.3±1.5 | 79.3±1.1 | 69.34 |
| GPT-GNNHu et al. (2020) | 64.5±1.4 | 74.9±0.3 | 62.5±0.4 | 58.1±0.3 | 58.3±5.2 | 75.9±2.3 | 65.2±2.1 | 77.9±3.2 | 67.16 |
| GraphLoGXu et al. (2021) | 67.8±1.9 | 75.1±1.0 | 62.4±0.2 | 59.5±1.5 | 65.3±3.2 | 73.6±1.2 | 73.7±0.9 | 80.2±3.5 | 69.70 |
| GraphCLYou et al. (2020) | 69.7±0.7 | 73.9±0.7 | 62.4±0.6 | 60.5±0.9 | 76.0±2.7 | 69.8±2.7 | 78.5±1.2 | 75.4±1.4 | 70.78 |
| JOAOYou et al. (2021) | 70.2±1.0 | 75.0±0.3 | 62.9±0.5 | 60.0±0.8 | 81.3±2.5 | 71.7±1.4 | 76.7±1.2 | 77.3±0.5 | 71.89 |
| JOAOv2You et al. (2021) | 71.4±0.9 | 74.3±0.6 | 63.2±0.5 | 60.5±0.7 | 81.0±1.6 | 73.7±1.0 | 77.5±1.2 | 75.5±1.3 | 72.14 |
| GraphMVPLiu et al. (2021) | 68.5±0.2 | 74.5±0.4 | 62.7±0.1 | 62.3±1.6 | 79.0±2.5 | 75.0±1.4 | 74.8±1.4 | 76.8±1.1 | 71.70 |
| MGSSLZhang et al. (2021) | 67.2±1.6 | 74.4±0.2 | 64.1±0.8 | 60.4±0.5 | 72.8±5.3 | 76.1±0.6 | 75.2±1.5 | 76.9±0.7 | 70.88 |
| GraphFPLuong & Singh (2024) | 72.2±0.7 | 73.9±0.2 | 63.5±1.0 | | 86.5±3.6 | 73.7±0.6 | 76.6±0.4 | 79.2±1.8 | 73.54 |
| HiMolZang et al. (2023) | 69.8±1.8 | 74.8±1.0 | 64.1±1.1 | 59.7±0.7 | 72.9±3.5 | 73.6±0.5 | 75.5±1.7 | 79.2±2.8 | 71.22 |
| DGPMYan et al. (2024) | 71.2±0.5 | 75.3±0.4 | 64.0±0.7 | 60.3±0.8 | 80.9±1.3 | 75.3±1.6 | 77.3±0.6 | 81.1±0.7 | 73.17 |
| MBMP (w.o. S) | 69.7±2.3 | 74.2±0.4 | 62.4±0.9 | 62.5±0.4 | 77.4±4.6 | 75.9±0.8 | 76.8±1.6 | 80.2±1.6 | 72.39 |
| MBMP (w.o. C) | 68.1±1.3 | 74.4±0.5 | 63.1±0.3 | 60.2±0.4 | 80.1±2.3 | 73.8±1.3 | 74.5±1.3 | 74.1±2.1 | 71.04 |
| MBMP (w.o. M) | 70.8±1.5 | 75.9±0.7 | 64.1±0.6 | 61.8±0.4 | 85.9±3.9 | 76.0±0.6 | 77.9±0.9 | 79.8±0.6 | 74.02 |
| MBMP (w.p. P) | 71.2±1.2 | 75.1±0.7 | 64.5±0.2 | 63.1±0.3 | 87.7±0.9 | 76.8±1.7 | 75.9±0.6 | 81.3±0.3 | 74.54 |
| MBMP | 72.9±0.7 | 74.7±0.3 | 64.4±0.7 | 61.6±0.6 | 89.1±1.5 | 79.1±0.9 | 78.4±1.0 | 81.5±1.2 | 75.20 |

the embeddings from both models during downstream prediction leads to substantial performance gains. Finally, a comparison across five categories shows that MBMP and its variants yield the best average performance. Motif-based pretraining methods follow, while contrastive learning methods and prediction or generation learning methods perform moderately well. Models without pretraining perform the worst. These results not only further demonstrate the effectiveness of our proposed approach, but also suggest that motif-based pretraining is more capable of capturing the local structures and higher-order connectivity of molecules.

### 4.3 IMPACT OF MOTIF EXTRACTION ALGORITHMS ON MOTIF-BASED PRETRAINING

To verify the effectiveness of our proposed graph autoencoder based motif extraction method (GAME), we compare it with existing motif extraction approaches by applying each to different motif-based pretraining methods. We exclude BRICS from the comparison as it lacks general applicability. Table 3 presents the experimental results, which demonstrate the effectiveness of our proposed GAME method. Among the four pretraining strategies, three achieve the best average ROC-AUC when combined with GAME, and it consistently yields superior performance across the majority of datasets. These results indicate that our proposed GAME method is more effective at capturing meaningful motif structures, thereby highlighting the importance of incorporating local atomic structural information in the motif extraction process.

### 4.4 PARAMETER ANALYSIS

To better assess the impact of the proposed Motif extraction algorithm (GAME) on model performance, we conduct a parameter analysis of GAME. Table 4 shows the results for the bond-breaking threshold $k$ and the minimum Motif frequency $N$. The experimental results show that the model performs best when a smaller value of $k$=5.5 is used. A smaller $k$ tends to produce overly large subgraphs, which can lead to an excessive number of low-frequency Motifs and negatively impact performance. In contrast, a larger $k$ results in overly fragmented subgraphs, making it difficult to capture the high-order molecular structures effectively. Additionally, for the minimum frequency $N$, the best performance is observed when $N = 50$. A smaller value like $N$=20 may include low-frequency Motifs that poorly represent local molecular structures. On the other hand, a large $N$ may filter out too many meaningful Motifs, retaining mostly smaller ones, which limits the ability to capture high-order connectivity within molecules.

Table 3: Test ROC-AUC on eight molecular property prediction datasets. The experimental results present the performance of four different motif-based pretraining methods under four distinct motif extraction algorithms. The **red** indicates the best performance under each pretraining method.

| Motif-Based Method | Motif Extraction Algorithm | BBBP | Tox21 | Toxcast | SIDER | ClinTox | MUV | HIV | BACE | Avg-AUC |
|---|---|---|---|---|---|---|---|---|---|---|
| MGSSL | JT-VAE | 66.1±1.1 | **74.6±0.2** | 63.6±0.8 | 61.3±0.6 | 71.2±3.3 | 77.3±2.4 | 75.1±0.3 | 75.9±2.5 | 70.65 |
| | MGSSL | 67.2±1.6 | 74.4±0.2 | **64.1±0.8** | 60.4±0.5 | 72.8±5.3 | **76.1±0.6** | 75.2±1.5 | **76.9±0.7** | 70.88 |
| | PS-VAE | 65.1±2.4 | 73.7±0.4 | 63.2±0.8 | 61.1±0.6 | **75.6±1.1** | 74.3±0.3 | 74.4±1.4 | 76.7±1.9 | 70.51 |
| | GAME(ours) | **68.2±1.9** | 74.5±1.6 | 63.8±0.4 | **61.5±0.3** | 72.4±1.4 | 74.6±1.9 | **75.3±0.8** | 75.6±1.5 | 70.74 |
| GraphFP | JT-VAE | 67.2±1.3 | 73.9±0.8 | **63.7±0.6** | **61.7±0.1** | 72.7±4.4 | 74.3±2.0 | 74.4±0.6 | 77.5±0.6 | 70.68 |
| | MGSSL | 68.5±1.2 | 73.0±0.9 | 62.2±0.5 | 61.1±0.7 | 65.6±3.7 | 73.9±1.5 | 74.5±0.9 | 77.7±1.5 | 69.56 |
| | PS-VAE | 68.3±1.5 | 74.2±1.2 | 63.2±0.7 | 59.8±0.8 | 74.2±1.2 | 73.3±3.6 | **76.1±1.4** | 78.5±2.0 | 70.95 |
| | GAME(ours) | **69.7±1.1** | **74.6±0.7** | 62.7±0.2 | 60.1±0.7 | **74.5±2.7** | **75.1±1.9** | 73.3±0.8 | **80.1±2.8** | **71.26** |
| HiMol | JT-VAE | 63.5±3.4 | 74.8±1.2 | 64.7±0.7 | 58.9±0.4 | 70.9±3.1 | 74.9±0.3 | 76.1±0.5 | **82.2±0.3** | 70.75 |
| | MGSSL | 68.2±1.1 | 74.2±0.1 | 65.3±1.8 | 59.7±0.2 | 70.5±1.9 | 73.3±1.8 | 76.4±0.1 | 80.8±0.5 | 71.05 |
| | PS-VAE | **68.6±1.3** | 75.5±0.1 | **65.4±0.4** | 59.2±0.5 | 71.5±2.6 | 73.5±2.1 | **77.6±1.0** | 80.4±0.5 | 71.46 |
| | GAME(ours) | **68.6±1.7** | **75.6±0.7** | 65.2±0.3 | **61.1±0.7** | **72.6±1.3** | **76.1±1.2** | 76.7±1.8 | 81.4±1.2 | **72.16** |
| MBMP(ours) | JT-VAE | 69.1±1.1 | 73.8±0.7 | 63.1±0.3 | 62.1±0.8 | 74.3±2.3 | 75.1±1.8 | 74.2±1.6 | 79.7±1.1 | 71.43 |
| | MGSSL | 69.2±1.0 | 74.0±0.7 | 61.8±0.8 | 61.6±0.7 | 67.7±2.9 | 75.3±1.2 | 75.1±1.4 | 78.1±1.7 | 70.35 |
| | PS-VAE | **69.9±1.2** | 73.6±0.9 | **63.6±0.7** | 61.1±0.8 | **78.9±2.2** | 75.3±1.8 | 73.9±1.1 | 77.7±2.3 | 71.75 |
| | GAME(ours) | 69.7±2.3 | **74.2±0.4** | 62.4±0.9 | **62.5±0.4** | 77.4±4.6 | **75.9±0.8** | **76.8±1.6** | **80.2±1.6** | **72.39** |

Table 4: Parameter analysis of bond-breaking threshold $k$ and minimum Motif frequency $N$.

| MBMP | BBBP | Tox21 | Toxcast | SIDER | ClinTox | MUV | HIV | BACE | Avg.AUC |
|---|---|---|---|---|---|---|---|---|---|
| $k = 0.50$, N = 50 | 72.2±1.3 | **75.5±0.6** | 64.2±0.8 | 61.2±0.8 | 87.1±3.5 | 78.8±0.8 | 77.6±2.1 | 80.2±2.4 | 74.60 |
| $k = 0.55$, N = 50 | **72.9±0.7** | 74.7±0.3 | 64.4±0.7 | 61.6±0.6 | **89.1±1.5** | **79.1±0.9** | 78.4±1.0 | **81.5±1.2** | 75.20 |
| $k = 0.60$, N = 50 | 70.8±1.1 | 74.9±0.6 | 63.7±0.6 | **62.8±1.3** | 87.3±2.2 | 78.4±2.7 | **78.7±0.5** | 80.8±2.1 | 74.68 |
| $k = 0.65$, N = 50 | 70.6±1.6 | 74.5±0.5 | **64.5±0.9** | 62.2±1.2 | 86.2±2.9 | 77.5±0.7 | 75.5±1.6 | 79.6±3.2 | 73.83 |
| $k = 0.55$, N = 20 | 71.7± 0.6 | 75.1± 0.3 | **65.1± 0.2** | **62.3± 1.3** | 85.2± 2.4 | 78.6± 1.2 | **79.1± 0.6** | 80.5± 1.8 | 74.70 |
| $k = 0.55$, N = 50 | **72.9±0.7** | 74.7±0.3 | 64.4±0.7 | 61.6±0.6 | **89.1±1.5** | **79.1±0.9** | 78.4±1.0 | **81.5±1.2** | **75.20** |
| $k = 0.55$, N = 100 | 70.7±1.3 | 74.9±0.5 | 64.2±0.2 | 61.8±1.3 | 87.4±1.2 | 77.6±0.8 | 77.8±0.9 | 80.2±2.7 | 74.33 |
| $k = 0.55$, N = 200 | 70.7±0.6 | **76.2±0.1** | 64.3±1.1 | 61.5±0.6 | 86.6±2.6 | 76.6±0.8 | 76.4±1.1 | 81.2±2.1 | 74.19 |

Furthermore, to analyze the impact of different loss components on model performance, we conduct a parameter study on the loss weight $\alpha$. The results in Table 5 show that performance peaks at $\alpha$=0.1, with performance first increasing and then decreasing as $\alpha$ grows. This indicates that both contrastive learning and matching learning from multiple perspectives are crucial for the model. Moreover, increasing the weight of prototype learning helps align Motif representations more closely, contributing to better pretraining.

Table 5: Parameter analysis of the loss weight $\alpha$.

| MBMP | BBBP | Tox21 | Toxcast | SIDER | ClinTox | MUV | HIV | BACE | Avg.AUC |
|---|---|---|---|---|---|---|---|---|---|
| $\alpha = 0.10$ | **72.9±0.7** | 74.7±0.3 | **64.4±0.7** | 61.6±0.6 | **89.1±1.5** | **79.1±0.9** | 78.4±1.0 | **81.5±1.2** | **75.20** |
| $\alpha = 0.20$ | 69.8±1.4 | 75.1±0.7 | 63.3±0.8 | 62.5±1.1 | 87.2±2.5 | 76.5±1.1 | 78.7±0.9 | 79.9±1.9 | 74.13 |
| $\alpha = 0.30$ | 70.9±0.8 | 75.0±0.7 | 64.1±0.6 | 61.9±0.9 | 88.7±0.8 | 77.5±0.7 | **79.2±0.3** | 80.2±1.5 | 74.69 |
| $\alpha = 0.40$ | 71.6±1.6 | 74.5±0.9 | 63.3±0.8 | **62.7±0.9** | 86.2±1.1 | 78.6±0.7 | 77.3±1.3 | 80.6±1.6 | 74.35 |
| $\alpha = 0.50$ | 71.5±0.8 | **75.3±0.3** | 63.9±0.9 | 61.9±0.7 | 85.7±0.1 | 76.7±1.1 | 78.1±0.5 | 80.9±0.8 | 74.25 |

## 5 CONCLUSION

To address the limitation of existing Motif extraction methods that overlook local atomic structure, we propose a novel graph autoencoder-based Motif extraction algorithm (GAME). By leveraging graph autoencoder-based structure reconstruction, GAME effectively identifies frequently occurring subgraphs, capturing the underlying Motifs of molecules. In addition, to better capture multi-perspective molecular information, we introduce a Motif-based multi-level molecular pretraining framework (MBMP). We pretrain our model on 250K unlabeled molecules from the ZINC15 dataset and evaluate it on eight widely used molecular benchmarks. Experimental results show that our model outperforms existing state-of-the-art methods. Furthermore, we compare our approach with existing Motif extraction and Motif-based pretraining methods, further demonstrating the effectiveness of both the proposed Motif extraction algorithm and the pretraining framework.

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
