# OpenReview forum: "Graph Autoencoder-based Motif Extraction Algorithm for Molecular Representation Learning"
_ICLR.cc/2026/Conference — Submitted to ICLR 2026_

### Official Review · Reviewer_hpJk · 2025-10-29

**Soundness:** 2
**Presentation:** 2
**Contribution:** 2
**Rating:** 2
**Confidence:** 4

**Summary:**

This study tackle the molecular representation learning problem and proposes a motif self-extraction method using graph autoencoder, which automatically identify frequently occurring local patterns. In addition, a motif-based multi-level molecular pretraining method is proposed to capture the local information and higher-order connections. Experiments were done on the 250K Zinc15 dataset to predict eight typical molecular properties.

**Strengths:**

* Motif-based contrast learning.

**Weaknesses:**

* There is considerable room for improving the writing to make the manuscript clearer and easier to understand.
* Section 3.2.2 is unclear. Please explain "permutation and concatenation".
* Section 3.2.3, what is the definition of a "motif class"?
* Table 2: The advantage of the proposed method over the baselines is not statistically significant in more than half of the prediction tasks.

**Questions:**

* Table 1: What does "Local-Struc" mean? Aren't motifs local structures? Methods for extracting motifs should consider local structure information.
* Line 85: Why need a graph autoencoder to evaluate the connection frequencies between atom pairs? Can this be simply done by counting? There are plenty of unsupervised data mining methods for discovering graph motifs. The author does not appear to be aware of them.

---

### Official Review · Reviewer_qKPK · 2025-10-30

**Soundness:** 2
**Presentation:** 2
**Contribution:** 2
**Rating:** 2
**Confidence:** 4

**Summary:**

This paper proposes GAME, a motif extraction algorithm that trains an autoencoder on structural reconstruction, which groups atoms with high-frequency connection patterns. The authors then introduce MBMP, a multi-level pretraining framework that uses these extracted motifs to capture both local and higher-order molecular information.

**Strengths:**

1. Motif extraction method is a crucial topic that needs more study in the community.
2. The paper is well-written and easy to follow.

**Weaknesses:**

1. My concern is that if the reconstruction task perfectly learns the identity function (making $F(\tilde{A})$ equal $A$, especially if  $\tilde{A}=A$, how can this same process also function as a filter to remove low-frequency edges and identify motifs?
2. What is a graph encoder used in the framework? A GNN or MLP?
3. I'm trying to understand the impact of the additional filtering step for low-frequency motifs. What proportion of motifs does this step remove, and how sensitive are the final results to this filtering? This is essential for accurately judging the effectiveness of the core motif extraction method.
4. The experimental results are not promising. The proposed method does not outperform other motif extraction methods.
5. Some motif-based molecular representation learning related works need to be discussed.

**Questions:**

Please refer to the weaknesses.

---

### Official Review · Reviewer_nN47 · 2025-10-31

**Soundness:** 2
**Presentation:** 2
**Contribution:** 2
**Rating:** 2
**Confidence:** 4

**Summary:**

The authors proposes GAME, a graph autoencoder-based motif extraction method that is trained to  perform structural reconstruction on molecular data. The authors claim that this is the first such GNN-based motif extraction method. They also introduce MBMP, a motif-based multi-level molecular pertaining framework for leveraging structural motif information in molecules via contrastive learning, cross-level matching learning, and prototype prediction learning. The paper further includes experiments on molecular property prediction datasets.

**Strengths:**

- I like the approach of incorporating several different losses in training to learn a more robust representation for a given molecule
- The experimental ablations for the various components of MBMP and the effect of different bond-breaking thresholds k and minimum motif frequencies N provides interesting additional insights into the method presented.

**Weaknesses:**

Major Weaknesses:
- There seem to be significant parts of the method that are missing or not presented properly in this work. A few examples are as follows:
    -  Line 194: “which uses its aggregation mechanism to capture the local structural pattern” - what is the aggregation mechanism? Depending on what aggregation is used, this has significant impacts on the expressive power of a given model
    - Line 197: “we use a decoder to calculate the probability of an edge” - again, what decoder do you use here?
    - Line 281: “MAP(·) is a mapping function that establishes the relationship between motifs and atoms” - this MAP function is not formally defined anywhere in the paper. What kind of mapping are you using?
- Why did the authors not compare against state of the art motif/hierarchical-based GNNs or graph transformer based architectures [1-6]? If one of the arguments is that this method can compete with domain-knowledge informed approaches, this is an essential comparison to show.
- All experiments are on the MoleculeNet dataset, which is known to have several flaws. Did the authors evaluate on any other molecular datasets, such as from ADME [7], OGB [8], LRGB [9], etc.? While all benchmarks have some issues to them, the experimental section of this paper would be greatly strengthened if success is shown across multiple datasets.

Minor Weaknesses:
- While the motivation of reducing reliance on chemical domain knowledge is somewhat sound, there are many cases where including it is greatly beneficial. Is there a way to incorporate this additional knowledge if desired?
- Although the paper is generally well laid out, it's somewhat sloppily written - there are several typos and misformated references present throughout the work that should be fixed and cleaned up.

Overall I believe this paper still requires significant work, and given the fact that there is no appendix or supplementary material, this submission seems too preliminary to be a full conference paper. Therefore, I recommend it for rejection.

References:
1. Bouritas et. al. Improving Graph Neural Network Expressivity via Subgraph Isomorphism Counting. In ICLR, 2021.
2. Barcelo et. al. Graph Neural Networks with Local Graph Parameters. In NeurIPS, 2021.
3. Bodar et. al. Weisfeiler and Lehman Go Cellular: CW Networks. In NeurIPS, 2021.
4. Jin et al. Homomorphism Counts for Graph Neural Networks. In ICML, 2024.
5. Luo et al. Enhancing Graph Transformers with Hierarchical Distance Structural Encoding. In NeurIPS, 2024.
6. Bao et al. Homomorphism Counts as Structural Encodings for Graph Learning. In ICLR, 2025.
7. Fang et al. Prospective validation of machine learning algorithms for absorption, distribution, metabolism, and excretion prediction: An industrial perspective. JCIM 2023.
8. Hu et al. Open graph benchmark: Datasets for machine learning on graphs. In NeurIPS 2020.
9. Dwivedi et al. Long Range Graph Benchmark. In NeurIPS 2022.

**Questions:**

See weaknesses.

---

### Official Review · Reviewer_GKrJ · 2025-10-31

**Soundness:** 3
**Presentation:** 4
**Contribution:** 3
**Rating:** 4
**Confidence:** 5

**Summary:**

This paper proposes a motif self-extraction method based on a graph autoencoder, as well as a motif-based pretraining method for molecular graph representation learning. First, assuming that frequently occurring substructural motifs in molecular tasks can be used in pretraining, the paper introduces a simple graph autoencoder to extract these motifs. By overlaying the adjacency matrices of the encoder-decoder output and the input graph and applying thresholding to remove low-value entries, the method fragments the molecular graph and builds a motif vocabulary. Based on this motif self-extraction, the paper proposes a pretraining strategy that combines three types of loss: motif-based contrastive learning, motif-to-atom level matching, and prototype learning of motif types. Experimental results show that this pretraining–fine-tuning approach outperforms benchmark methods.

**Strengths:**

- This paper presents a simple but quite effective "self-extraction" method for motif vocabulary building. The idea of leveraging the output of a graph autoencoder to effectively fragment molecular structures for motif discovery is itself a very interesting concept.
- Building on this motif extraction approach, the paper proposes a motif-aware pretraining strategy for learning molecular representation, and demonstrates its effectiveness through experimental comparisons. This reflects recent insights suggesting that incorporating motif-level knowledge is quite effective for molecular representation learning, making it a noteworthy contribution.

**Weaknesses:**

- While the main focus of the paper is on pretraining, it's unfortunate that there's no analysis comparing the extracted motifs to those obtained using conventional methods such as BRICS since BRICS is also based on fragmentation of the molecular graphs. Understanding how similar or different these motifs are would have added valuable insight.
- The paper proposes both a motif extraction method and a corresponding pretraining strategy, but an ablation study is essential to verify the effectiveness of each component individually.
- For instance, MGSSL (Zhang et al., 2021), which is cited in the paper, uses BRICS as its motif vocabulary. Therefore, it’s important to investigate whether simply replacing BRICS with the proposed self-extracted motifs could lead to improved performance. Similarly, for other motif-aware approaches that rely on explicit motif vocabularies, it would be necessary to test whether swapping in the proposed method yields benefits.
- Conversely, the proposed pretraining strategy should, in principle, be applicable to existing motifs as well. This raises the question of how effective it would be when used with BRICS motifs, for example. As I understand it, Eq. (15) does not incorporate the loss used for vocabulary construction in Eq. (3), meaning that vocabulary building is treated separately from pretraining and is not part of an end-to-end framework.
- The individual components of the pretraining strategy are conceptually sound, but they appear incremental and lack novelty. A clearer explanation of how this approach differs from existing methods such as MGSSL is essential.

**Questions:**

- Have you investigated how the motifs extracted by your proposed graph autoencoder-based method compare to those obtained through established motif extraction approaches, such as BRICS in terms of similarity or differences?
- In existing motif-based pretraining approaches, such as MGSSL (Zhang et al., 2021), have you investigated how replacing the BRICS motif vocabulary with the one obtained through your proposed graph autoencoder-based self-extraction affects prediction accuracy?
- In your proposed pretraining strategy, have you examined how using existing vocabulary building strategies such as BRICS affects prediction accuracy?

---

### Meta-Review · Area_Chair_uupd · 2026-01-04

**Summary:**

Contribution Summary:

The paper proposes GAME, a graph autoencoder-based method for self-extracting molecular motifs, and MBMP, a pretraining framework leveraging these motifs via contrastive learning. The authors claim this reduces reliance on domain knowledge compared to rule-based methods.

Reviewer Concerns:
* **Methodological Ambiguity:** Multiple reviewers pointed out severe gaps in the technical description, including undefined aggregation mechanisms, decoders, and mapping functions, making the work unreproducible and difficult to assess.
* **Conceptual Validity:** Reviewers questioned the fundamental premise, asking why a complex graph autoencoder is necessary over simple counting or existing rule-based methods like BRICS, and how an autoencoder trained for reconstruction can effectively act as a frequency filter.
* **Experimental Rigor:** The evaluation was criticized for relying solely on MoleculeNet (ignoring OGB/ADME), lacking statistical significance in results, and failing to compare against state-of-the-art graph transformers or established motif extraction baselines.

**Reviewer Concerns:**

The authors failed to submit a rebuttal or respond to any reviewer inquiries, leaving all technical and experimental concerns unaddressed.

Outstanding Concerns:
* **Missing Baselines:** Reviewer GKrJ and hpJk correctly noted the absence of comparisons to standard motif extraction methods (e.g., BRICS) or simple data mining approaches. Reviewer nN47 noted the lack of SOTA GNN/Graph Transformer baselines.
* **Undefined Methodology:** Reviewer nN47 highlighted that key components (aggregation, decoder, MAP function) were not formally defined. Reviewer hpJk found Sections 3.2.2 and 3.2.3 unclear.
* **Questionable Efficacy:** Reviewer qKPK raised a logical concern: if the autoencoder learns the identity function perfectly, it cannot function as the proposed filter. This core conceptual flaw remains unaddressed.
* **Statistical Significance:** Reviewer hpJk noted that the proposed method's advantage is not statistically significant in more than half of the tasks.

**Reviewer Scores:**

* **Reviewer GKrJ:** Original Score: 4. Final Score: 4 (Reject).
    The reviewer was the most positive but still recommended rejection due to missing ablations and comparisons to BRICS. The lack of response confirms the paper is not ready.
* **Reviewer nN47:** Original Score: 2. Final Score: 2 (Reject).
    The reviewer identified significant missing methodological details and cited specific missing benchmarks (OGB, LRGB). Without a rewrite and new experiments, the score stands.
* **Reviewer qKPK:** Original Score: 2. Final Score: 2 (Reject).
    The reviewer questioned the fundamental logic of the autoencoder design. Without an explanation, the method appears theoretically unsound.
* **Reviewer hpJk:** Original Score: 2. Final Score: 2 (Reject).
    The reviewer pointed out the lack of statistical significance and questioned the necessity of the complex approach over simple counting.

---

### Decision · Program_Chairs · 2026-01-26

Reject